

# Possibilistic response surfaces combining fuzzy targets and hydro-climatic uncertainty in flood vulnerability assessment

Thibaut Lachaut[1] and Amaury Tilmant[1]

[1]Laval University, Québec, QC G1V 0A6, Canada

**Correspondence:** Thibaut Lachaut (thibaut.lachaut.1@ulaval.ca)

**Abstract.** Several alternatives have been proposed to shift the paradigms of water management under uncertainty from predictive to decision-centric. An often mentioned tool is the stress-test response surface; mapping system performance to a large sample of future hydro-climatic conditions. Dividing this exposure space between success and failure requires clear performance targets. In practice, however, stakeholders and decision-makers may be confronted with ambiguous objectives for which there are no clearly-defined (crisp) performance thresholds. Furthermore, response surfaces can be non-deterministic,

as they do not fully capture all possible sources of hydro-climatic uncertainty. The challenge is thus to combine two different types of uncertainty: the irreducible uncertainty of the response itself relative to the variables that describe change, and the fuzziness of the performance target. We propose possibilistic surfaces to assess flood vulnerability with fuzzy performance thresholds. Three approaches are tested and compared on a un-gridded sample of the exposure space: (i) an aggregation of logistic regressions based on $\alpha$-cuts combines the uncertainty of the response itself and the ambiguity of the target within a single

possibility measure; (ii) an alternative approximates the response with a fuzzy analytical surface; and (iii) a convex delineation expresses the largest range of failure specific to a given management rule without probabilistic assumptions. To illustrate the proposed approaches, we use the flood-prone reservoir system of the Upper Saint-François River Basin in Canada as a case study. This study shows that ambiguity can be effectively be considered when generating a response surface and suggests how

further research could build a possibilistic framework for hydro-climatic uncertainty.

## 1 Introduction

Uncertainty is a driving force of the transformations of human societies. Settlement, agriculture and animal husbandry, storage facilities, many innovations seek to make the future more predictable, and thus allow for investments with a better knowledge of associated risks. For some social scientists, uncertainty and risk become more central, even constitutive of society, following

the industrial revolutions (U. Beck, 1985). As the scope of human activity expands exponentially and meets the boundaries of its functional environment (J. Rockström et al. 2009), the adverse externalities shape a new layer of human-induced risks. Nuclear and chemical catastrophes were the first to attract global attention, later joined by the perturbation of climatic and biophysical mechanisms at a global scale.

     The relationship between society and water follows the same path. Five millennia of engineering have seen the development

of reservoirs, irrigation, levees and aqueducts in order to counter the uncertainty inherent to precipitations and river flows. The





water domain is consequently confronted to the same new layer of uncertainty and risks stemming from human externalities. Climate change is a looming threat on current investment or planning decisions in water resources (IPCC, 2014), while increased pressure at the basin level makes hazards, appropriations and conflicts all the more impactful (Srinivasan et al., 2012), as illustrated by the closed basin concept (Molle et al., 2010).

The way water management - both science and practice - handles uncertainty is crucial. Not only does uncertainty justify intervention and planning, but the way decisions are taken is also based on different interpretations of uncertainty. The dominant paradigm has been to optimize investments or management plans according to the most probable future, a knowledge based on the statistical analysis of historical time series and assuming their stationarity. This assumption of stationarity has been contested however as anthropogenic activities do affect the very climatic processes that led to past hydrologic behavior (Milly

et al., 2008).

Water planning can also rely on the modeling of future scenarios and climate change impacts over existing hydro-climatic conditions. This widely used method relies on General Circulation Models (GCM) that simulates future global climates depending on assumptions (Representative Concentration Pathways, RCP) about the $CO_2$ concentrations in the atmosphere or more generally the radiative forcing (Brown and Wilby, 2012, Weaver et al, 2013). Results from global simulations are translated

into local hydro-climatic projections through a downscaling process. Hydrological modelling then translates climatic variables into run-off time series. Such an approach has its own limitations however. $CO_2$ emission pathways depend on worldwide future policy choices which are not yet known nor even predictable. Moreover, climate models carry their own structural uncertainties, and so are the downscaling processes (Prudhomme et al., 2010, Mastrandrea et al., 2010, Kay, et al., 2013, Weaver et al., 2013, Kim et al., 2019). Besides, a discrete set of projections is not suited to find the hydro-climatic thresholds beyond

which a system fails to reach its target (Culley et al., 2016). Such a risk assessment process is also increasingly unreliable with systems that operate with shorter time steps and extreme events, like flood control operations (Knighton et al., 2017).

In the last 15 years there has thus been a widespread effort to find new paradigms to make decisions under deep uncertainty, notably through a greater focus on the decision process rather than on improving predictions (Lempert et al., 2006, Maier et al., 2016, Lempert, 2019). Switching to a robust or decision-centric paradigm always seeks to increase the sampling of

hydro-climatic conditions, and relies on a sensitivity analysis of a water system to driving variables rather than evaluating the consequences of the most probable future and optimizing accordingly (Weaver et al., 2013). A consolidation of the field is proposed under the decision making under deep uncertainty (DMDU) denomination (Marchau et al., 2019).

One of the most common tools within the decision-centric framework is the response function or surface (Prudhomme et al., 2010, Brown et al., 2012, Culley et. al., 2016, Brown et al., 2019, Nazemi et al., 2020). Through a stress-test, "bottom-up"

approach, a water system is simulated for a large set of conditions representing possible evolutions of some uncertain hydro-climatic variables (or stressors), establishing a relationship between such stressors and the performance of the system. Such an approach is sometimes called scenario-neutral (Prudhomme et al., 2010, Broderick et al., 2019) as it doesn't intrinsically rely on GCM outputs and RCP assumptions. Alternatives, like making new investments, changing management schemes, are compared through their respective performance outcome over a whole space of possibilities, or exposure space (Culley et al.,

2016). In the Decision Scaling approach (Brown et al., 2012, Brown et al., 2019) GCM projections can then be introduced





as weights on the response surface to inform probabilities associated to climate states. GCMs can thus remain useful without conditioning the decision process, and once updated their outcomes on the system can be mapped on the response without the need for new simulations of the water system. The intention shared within the overall decision-centric framework is to adapt classic risk assessment to the "death of stationarity" (Milly et al., 2008) while producing information more useful and engaging than a fully descriptive scenario approach (Weaver 2013). Response surfaces have been illustrated by many case studies (e.g. Nazemi et al., 2013, Turner et al., 2014, Whateley et al., 2014, Herman et al., 2015, Steinschneider et al., 2015, Spence et al., 2016, Pirttioja et al., 2019, Ray et al., 2020), expanded to many-objectives or stakeholder systems (Poff et al., 2016; Culley et al., 2016, Kim et al., 2019) and sometimes officially adopted in management processes (Moody and Brown, 2013, Weaver et al., 2013, Brown et al., 2019).

Although the response surface is a powerful and efficient tool to circumvent the problems and arbitrariness brought by "top-down", GCM-based assessments, the applications to date remain relatively recent and scarce (Guo et al., 2018). Moreover, many assumptions associated with the stress test approach can introduce additional uncertainty.

One source can be the ambiguity of the user-defined performance targets (Maier et al., 2016). The stress-test approach needs performance target values (thresholds) in order to separate the exposure space between accepted and rejected domains. However such targets are often unclear or arbitrary; and are heavily reliant on political, sociological and institutional processes (El-Baroudy and Simonovic, 2004). Fuzzy set theory (Zadeh, 1965) provides an analytical framework to characterize and manipulate stakeholders' ambiguity (Huynh et al., 2007). It has been extensively used in the water domain (El-Baroudy and Simonovic, 2004, Qiu et al., 2018) in particular to solve multi-objective decision-making problems (e.g. Jun et al., 2013). However, to the best of our knowledge, fuzzy set theory has not yet been used to handle imprecise thresholds between satisfactory and failure regions of a response surface. The very notion of an arbitrary threshold defining success, like flood control reliability above 0.95, can be considered as a departure from a strictly probabilistic framework and could justify a complementary possibilistic approach based on fuzzy sets (Dubois et al., 2004).

Independently from performance targets, response functions also have their own noise or internal uncertainty, as their selected driving variables can only partially explain hydrological and climatic uncertainties. As such, performance is an expected value rather than a deterministic one, hence possibly underestimating real risks. Irreducible uncertainty usually requires adaptive management (Brown et al., 2011), but there is interest to integrate part of this information into the response surface tool. Kay et al. (2014) proposed the use of uncertainty allowances that could vary depending on the response type and catchment. More specifically, flood control systems operate on shorter time scales and are even harder to assess over long term climate shifts (Knighton et al. 2017), thus also more challenging to evaluate with response functions. Kim et al. (2018) stress how the choice of modelling time scale (daily vs hourly) can lead to risk underestimation. The choice of different scenario-neutral methods can lead to different results (Keller et al., 2019), notably the choice of the synthetic series generator (Nazemi et al., 2020). Steinschneider et al. (2015) compare different sources of uncertainty, acknowledging the strong impacts of hydrological modelling and internal climate variability compared to long term climate uncertainty, as well as Whateley and Brown (2016). Testing a limited number of stressors as explaining variables therefore leads to a response function that returns uncertain performance. Kim et al. (2019) propose to associate probabilities to uncertain response functions through logistic regression, while





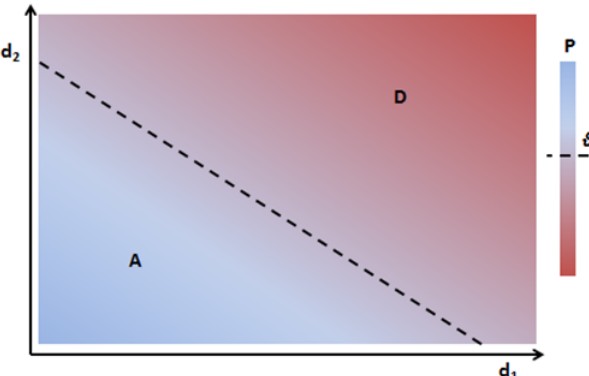

**Figure 1.** Concept of the response surface as a stress-test with describing variables $(x_1, x_2)$. Success and failure regions are defined by a threshold $\theta$ over performance $p$.

Tanner et al. (2019) do so with a Bayesian belief network model. It is here noted that most response surfaces follow gridded sampling, which can also be a loss of information (e.g. Huang, 2000 for elevation models) and thus risk under-estimation.

The objective of the present study is to combine with a possibilistic approach two different types of uncertainty: the fuzziness of performance targets and the irreducible uncertainty of the response surface. The rationale behind it and three tested implementations are presented in section 2: a numerical approximation of a fuzzy-random logistic regression, a fuzzy analytical approximation of the response itself and a convex delineation of the largest range of failure. A case study is presented in section 3, a flood-prone reservoir system in southern Québec, Canada. Results are presented in part 4, followed by a discussion on the respective merits and limitations of the proposed methods.

## 2 Methods

### 2.1 Rationale

#### 2.1.1 Uncertain response function

We first consider how a limited set of variables leads to an inherently uncertain response function, and how it relates to the partition of the exposure space and the decision process.

A stress test consists in assessing the performance of a system for a large enough number of situations, in order to identify *which of these situations* leads to an unsatisfying performance, or overall failure.

Often inspired from Hashimoto et al. (1982), a performance indicator is a statistical measure of local failure duration or amplitude, aggregated over a certain time period. Local failure is a state of the system at a given time step: a state of flooding can be defined by a streamflow exceeding a threshold at any moment. The overall performance of the system over a period



quantifies its ability to mitigate the number or amplitude of local failures. For example, the reliability of a flood control system
can be measured as the proportion of a given period where no flooding happens. When performing a stress-test of a system,
overall success or failure is usually defined by a performance target $\theta$, for example reliability above 0.95 over a given period
can define overall success.

A stress-test maps the performance p on a response surface, to a limited number of descriptive variables $x_i$. It aims at
delineating the subsets $A$ and $D$ of overall success and failure (Fig. 1). Such variables, like the mean flow, the peak flow, or
temporal autocorrelations, are aggregations of the time series that are the inputs of a water system simulation. Because a limited
number of descriptors do not capture all possible fluctuations of a time series, a term of irreducible uncertainty remains. The
response surface is then given by:

$$p = g(x_1, x_2...) + R \tag{1}$$

In a risk-averse approach, the objective is to find the range of failure (more than success), the space over which a system
fails to satisfy a performance target $\theta$. With 2 variables, this space is the set of solutions $D = (x_1^*, x_2^*)$ to the inequation $p < \theta$,
so

$$g(x_1, x_2) + R < \theta \tag{2}$$

Simplifying the response surface by averaging it over $p$ (*vertically*) can thus under-estimate the failure domain. Irreducible
uncertainty can be addressed through adaptive management (Brown et al., 2011), uncertainty allowances (Kay et al., 2014), and
extensive Monte-Carlo sampling (Whateley and Brown, 2016). If possible though, it can be convenient to directly integrate
information about remaining uncertainty within the response surface itself. It can be represented through a transition zone
between success and failure domains, as performed by Kim et al. (2019) with a logistic regression. Besides, most studies use
gridded sampling of the exposure space, which is a *horizontal* aggregation that also results in information loss like in the case
of digital elevation models (Huang, 2000), and which in this case can also under-estimate risks. A simple un-gridded alternative
is proposed in section 3.2.

### 2.1.2 Fuzzy performance targets

The performance target $\theta$ defines the set of successful outcomes. It is a subjective or arbitrary opinion from stakeholders or
decision makers to attribute a normative value to a certain performance level. The vast majority of the studies reported in
the literature assume that the threshold between satisfactory and unsatisfactory states is crisp (Brown et al., 2012, Culley et
al., 2016, Kim et al., 2019). As such a threshold shapes directly the partition of the response function, with a crisp value the
exposure space can be subdivided in only two sub-spaces: failure versus success.

The very existence of a target is the basis of satisficing behaviors (Simon, 1955) that differ from utility maximizing behaviors
as coined by Von Neumann and Morgenstern (1944). In practice however, while clearly following a satisficing model, there
might be situations whereby the water manager is unable (or unwilling) to provide a crisp, well-defined target, or when such
threshold is disagreed upon by stakeholders. For example, when controlling water levels in a reservoir to prevent inundations,

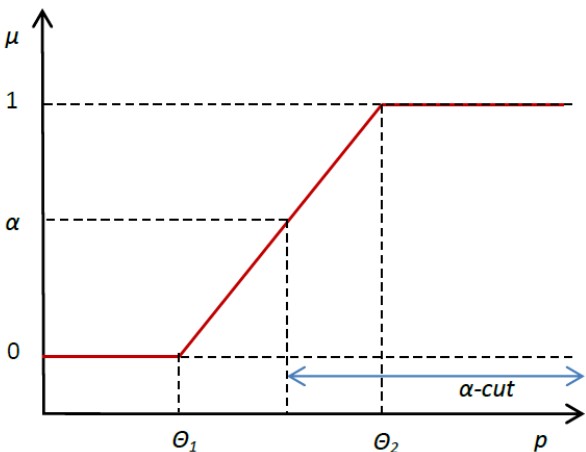

**Figure 2.** Concept for a fuzzy set of success $A_\mu$ over performance $p$.

the operator can handle certain tolerances above the maximum desired level. Of course, the greater the deviation from the desired level, the less acceptable it becomes.

Mathematically, fuzzy sets theory (FST) handles imprecisely-defined or ambiguous quantities. Introduced by Zadeh in 1965, fuzzy sets theory has become a common tool in decision making analysis or computational sciences when non-probabilistic

uncertainty stemming from ambiguity or vagueness must be considered (Yu et al., 2002). In our case, FST allows us to introduce vagueness in target-based decision making, without forsaking a target-based model in favor of an unbounded maximizing behavior (although a fuzzy target can also be seen as a generalization of both maximizing and satisficing behaviors – see Castagnoli and LiCalzi, 1996, and Huynh et al., 2007).

We consider here the case where such a target $\theta$ may not be precisely defined by stakeholders but can take many subjective

qualifications from acceptable to unbearable, hence relaxing (without fully removing) the arbitrary condition of satisfying a crisp value. A fuzzy set $A_\mu$ of acceptable states therefore qualifies the performance $p$ with a membership value comprised between 0 and 1. The membership function $\mu$ associated to the fuzzy set $A$ describes the degree to which any value of $p$ more or less belongs to $A$ (Figure 2, eq 3).

$$\begin{cases} \mu(p) = 0 & p < \theta_1 \\ 0 < \mu(p) < 1 & \theta_1 \geq p < \theta_2 \\ \mu(p) = 1 & p \geq \theta_2 \end{cases} \tag{3}$$

When a threshold corresponds to a fuzzy set, it means that there is a transition zone between success and failure states where intermediate levels of membership exist. Conversely, another interpretation is that the membership function is the distribution of the *possibilities* (Zadeh 1978, Dubois and Prade, 1988) that any given performance $p$ represents a success.





An $\alpha$-cut $A_\alpha$ is the crisp set over $A_\mu$ for which the membership degree to $A_\mu$ is equal or above $\alpha$. The largest $\alpha$-cut is called the *support* of the fuzzy set $A_\mu$ ($p \geq \theta_1$). The smallest $\alpha$-cut is the *core* of the fuzzy set ($p \geq \theta_2$).

$$A_\alpha = \{p \in A_\mu \mid \mu(p) \geq \alpha\} \tag{4}$$

## 2.2 Combination of fuzzy targets and uncertain response function

The challenge is to combine two different sources of uncertainty described in section 2.1: the uncertainty or low quality of the response itself relative to the variables that describe change, and the fuzziness of the performance target. In order to integrate both, three methods are suggested: an approximated fuzzy-random logistic regression, an analytical approximation of the response surface and a convex delineation of the space of failure.

### 2.2.1 Approximation of a fuzzy-random logistic regression

As the goal of the response surface is to divide the exposure space between success and failure, the value associated to any combination of variables can be either 0 or 1 if a specific performance target $\theta$ is reached or not. As seen in section 2.1, an intrinsic uncertainty remains in response surfaces. Kim et al. (2019) introduce the logistic regression to incorporate probabilistic information into the response surface. The logistic regression is used to explain a binary outcome from independent variables $(x_1, x_2)$, and returns a probability of success $\pi$ :

$$\pi_\theta = \frac{1}{1 + \exp\left(-(\beta_0 + \beta_1 x_1 + \beta_2 x_2 + \dots)\right)} \tag{5}$$

$$\pi_\theta(x_1, x_2) = P(p \geq \theta) \tag{6}$$

where $x_i$ are the defining variables of the exposure space and $\beta_i$ the regression coefficients. The logistic response surface therefore provides the probability $\pi$ of meeting the target $\theta$ over the $(x_1, x_2)$ exposure space. Partitions of the space between success and failure sub-spaces, that can be defined as $\pi - cuts$, are now relative to a specific probability of success $\pi^*$ taken by $\pi_\theta$:

$$S_{\pi^*} = \{x_1, x_2 \mid \pi(x_1, x_2) \geq \pi^*\} \tag{7}$$

By considering the domain of successful outcomes as a fuzzy set, we introduce a layer of uncertainty that is different in nature from the irreducible hydro-climatic uncertainty. While the logistic regression returns a *probability* of surpassing any given performance target for a combination of variables (eq. 5 and 6), the fuzzy set of success returns the *possibility* of any such performance target being actually considered as a success (eq. 7).

Fuzzy regression models, including fuzzy logistic regression (e.g. Pourahmad et al., 2011, Namdari et al., 2014) replace probabilities by fuzzy numbers; they usually do not combine them. Fuzzy probabilities (Zadeh, 1984) are considered within the so-called fuzzy random regression field, however no fuzzy random logistic regression seems to have been developed to date (see Chukhrova and Johannssen, 2019, for a review of the fuzzy regression field).



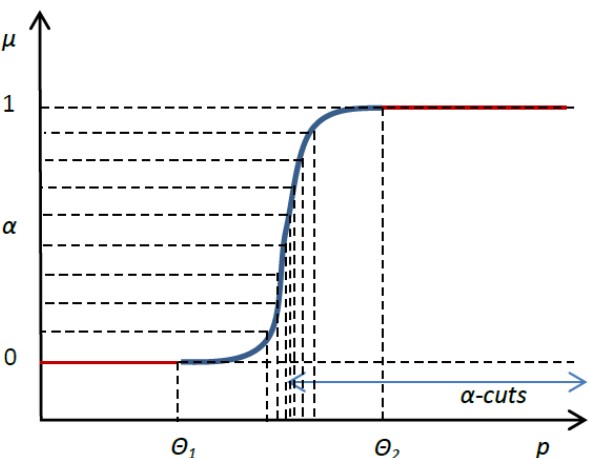

**Figure 3.** Concept for $\alpha$-cut sampling, sigmoid function

Here we use a discretised approximation of a fuzzy random logistic regression based on $\alpha$-cuts. A single target $\theta$ defining a crisp set of success $A$ is also an $\alpha$-cut of the fuzzy set of success $A_\mu$. Then a single logistic regression for any success threshold $\theta$ is also the probability of belonging to the $\alpha$-cut of the fuzzy set of success defined by $\theta$:

$$\pi_\theta\left(x_1, x_2\right) = P\left(p \in A_\alpha\right) = P\left(p \in A_\mu \mid \mu\left(p\right) \geq \alpha\right) \tag{8}$$

with $\alpha = \mu(\theta)$.

Following the interpretation of Huynh et al. (2007), the overall possibility $\Pi$ of the random variable $p$ belonging to the fuzzy set $A_\mu$ can be given by the integral over $\alpha$ of the probabilities of success defined at every $\alpha$-cut.

$$\Pi\left(x_1, x_2\right) = P\left(p \in A_\mu\right) = \int_0^1 P\left(p \in A_\mu \mid \mu\left(p\right) \geq \alpha\right) d\alpha \tag{9}$$

And thus

$$\Pi\left(x_1, x_2\right) = \int_0^1 \pi_{\mu^{-1}(\alpha)}\left(x_1, x_2\right) d\alpha \tag{10}$$

The approximated logistic regression for a fuzzy set of success is therefore the average of the logistic regressions for all the associated $\alpha$-cuts. With a uniform discretization of 10 alpha levels, the spacing of every $\alpha$-cut, defined with $\theta = \mu^{-1}(\alpha)$, relies on the shape of the membership function. A linear shape of $\mu(p)$ leads to a uniform sampling of the $\alpha$-cuts, while a sigmoid shape leads to a Gaussian sampling of $\alpha$-cuts centered on $\theta$ (Fig. 3).



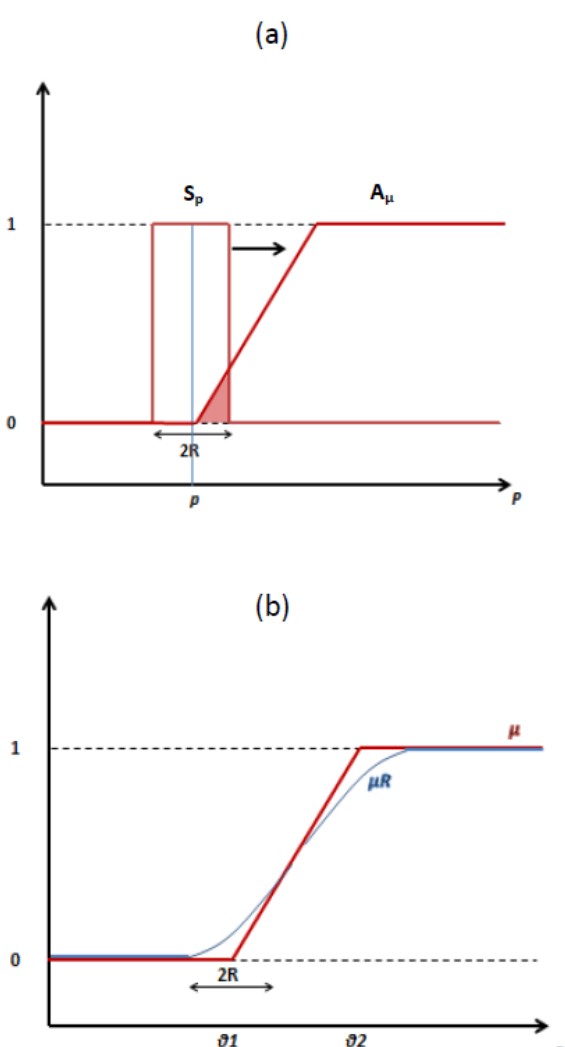

**Figure 4.** (a) Fuzzy set $A_\mu$. (b) Fitting error set Sp. (c) Intersection between $A_\mu$ and $S_p$ (shaded area). (d) resulting membership function $\mu_R$.

### 2.2.2   Analytical approximation of the response function and fuzzy set intersection

Instead of fitting a logistic function to the binary outcomes of success and failure, performance itself can be directly approximated as an analytical response surface (eq. 3). The final outcome is then a direct mapping from the performance approximation to a [0 1] degree of success with the membership function of the fuzzy set of success $A_\mu$ (eq. 5). For every $(x_1, x_2)$, a single approximated performance is given by $p^* = g^*(x_1, x_2)$, so the possibilistic response surface is defined by $\mu_R(p^*)$. The membership function $\mu$ is modified as follows to account for the fitting error $R$.





To any value of $p^*$ is associated the membership degree $\mu$ between 0 and 1 depending on $\theta_1$ and $\theta_2$, 1 defining complete success. The fitting error, or uncertainty around performance $p^*$ can be expressed as another set $S_p$, centered on any value p

with a 2R-sized support (Figure 4a). This set can also follow any shape depending on the user's risk aversion. With a risk-averse attitude, a crisp set defines here R assuming a uniform possibility distribution. But an actual distribution of R around the approximation could also be used.

The modified membership degree $\mu_R(p^*)$ should account for the 2R-large interval that represents the possibility domain around $p^*$. So at any given $p^*$, the possible acceptability values are represented by the intersection between the sets $S_p$ and $A_\mu$,

given by the MIN operator. The resulting value $\mu_R(p^*)$ is the average over this intersection.

The new membership function $\mu_R$ over the entire domain of performance is then the moving average of $\mu$ with 2R window size (Figure 4b). A single possibility surface is thus obtained for any $(x_1, x_2)$ coordinate (eq. 11).

$$\mu_R(x_1, x_2) = \mu_R(p^*) = \frac{1}{2R} \int\limits_{p^*-R}^{p^*+R} \mu\ (p^*)\, dp^* \tag{11}$$

### 2.2.3 Convex hulls as range of success and failure

The climate stress test seeks to identify accepted and rejected sub-spaces A and D within the exposure space. As seen in section 2.1.1, gridded sampling can result in risk under-estimation. With an uncertain, noisy response function and a non-gridded sampling of the exposure space, the sets A and D of accepted and rejected points do not form two cohesive, identifiable and mutually exclusive sub-spaces. The methods described in sections 2.2.1 and 2.2.2 are regression or surface fits that incorporate the remaining errors but are still approximations and might not represent all possibilities of success or failure. In a risk-averse

approach, decision-relevant outliers could also be considered, in order to prepare for the most unlikely, but possible failures. The question is which performance values should be attributed at any location between sampled points.

One simple way to conservatively identify a sub-space from a set of points is their convex hull, the smallest possible convex space that contains the set. Convex hulls are extensively used in point process analysis and notably decision theory and risk analysis (Harris 1971). They can be used to identify failure regions when a response surface is inadequate, e.g. in mechanical

engineering (Missoum et al., 2007).

The underlying assumption is that, for any triangle of points contained in a set, any point within the triangle also belongs to the set. Following further a possibility-centric approach, what is sought here is the largest convex range of failure (LCRF). While less impactful for a risk-averse decision process, the largest range of success can also be expressed to further differentiate the regions of the response function. With a deterministic response function a single threshold will discriminate a space between

two complementary sub-spaces, accepted and rejected. With a noisy response and a crisp target, both sub-spaces will overlap, creating a transition zone (Fig. 5).

Considering a fuzzy performance target we modify the definition of both accepted and rejected sub-spaces. The loosely accepted sub-space is the convex hull of all performance values *superior to the lower threshold*. The loosely rejected sub-space is the convex hull of all performance values *inferior to the higher threshold*.





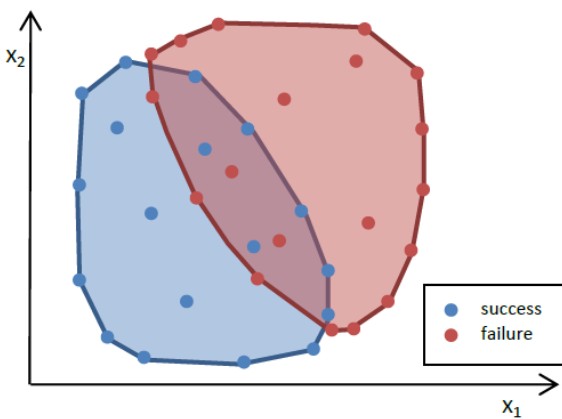

**Figure 5.** Concept for largest convex ranges of success and failure

Said otherwise, because only the upper bound of possibility is sought for success and failure, the largest range of failure (resp. success) is simply defined by *the smallest convex set of points where p belongs to the smallest (resp. largest) $\alpha$-cut of $A_{\mu}$ also called core (resp. support).*

    This method gives more weight to outliers, as they define the convex hull. It is a simple measure of possibility, and does not discriminate points within the transition zone. Different management rules are compared according to the relative position and 250  overlap areas of their respective transition zones.

    As decision-centric methods rely on large number of simulations, computing power parsimony is an applicability concern (e.g. Whateley et al., 2016, Zatarain Salazar et al., 2017). The LRCF is only defined by its vertices, and only the border closer to the success region matters for decision purposes. Large parts of the response surface could potentially be ignored, saving computation time. An iterative sampling of the response function can thus complement the LCRF method.

## 255  3   Application

A reservoir system in eastern Canada is used as case study to illustrate the applicability of the possibilistic response surfaces

### 3.1   Upper Saint-François River Basin features

The Upper Saint-François River Basin (USFRB) is located in the province of Quebec, Canada. The selected gauging point, near the agglomeration of Weedon, drains an area of 2940 km$^2$ with an average annual flow of 2.1 billion cubic meters. The 260  system (Fig. 6) involves the Saint François River, controlled by two reservoirs Lake Saint-François and Lake Aylmer with a combined storage capacity of 941 million cubic meters, and the uncontrolled affluent Saumon River.

    Both reservoirs are managed by the Ministry of Environment and Fight against Climate Change through the CEHQ water agency. The main operational objectives are: (i) to protect the municipality of Weedon and several residential areas around the





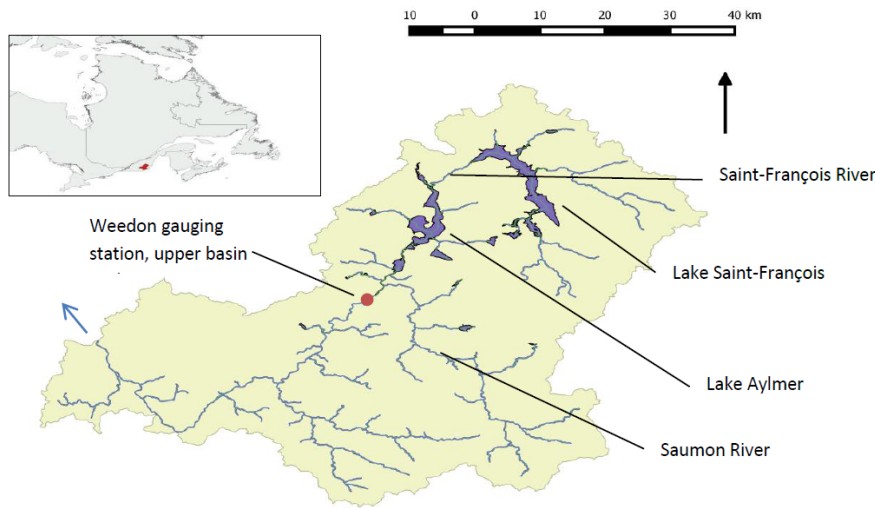

**Figure 6.** Layout of the Upper Saint-François River Basin, Québec, Canada.

lakes from floods, (ii) to ensure minimum river discharges and water levels in the lakes to preserve aquatic ecosystems, (iii) to

provide the downstream run off river power station with a reliable water discharge; and (iv) to maintain desired water levels in the lakes for recreational uses during the summer.

This multipurpose reservoir system thus follows a refill-drawdown cycle accordingly. With a snowmelt dominated flow regime, the reservoirs are emptied in winter, filled in spring and aim at a stable level in summer. Difficult snowpack volume estimation and variable precipitations can lead to under or over-estimation of the spring flood, leading to either insufficient

summer levels or frequent flooding of the downstream agglomeration of Weedon as experienced in 2018 and 2019.

### 3.2 Inflow time series

Following Nazemi et al. (2013), Borgomeo et al. (2015), Herman et al. (2016), Zeng et al. (2017) or Nazemi et al. (2020), we use streamflow stressors instead of climatic ones, as the present study does not aim at differentiating between several sources of uncertainty – climate change, climate variability, run-off modelling – but proposes a method that accommodates any of them.

In Québec, the CEHQ water agency regularly produces projections of river discharges throughout the province as part of the Hydroclimatic Atlas based on climate models (see Hydro-climatic Atlas, 2015, 2017). In this study, available time series (MELCC, 2018) can be used as basis for synthetic streamflow generation, then directly plotted on a response function according to their own $(x_1, x_2)$ coordinates, filling the response function without following a gridded sampling. Besides avoiding the spatial simplification entailed by gridded sampling (section 2.1.1.), such a method makes a greater use of hydro-climatic future

scenarios when many are already available, providing an already high diversity of times series (different GCM simulations, RCP scenarios and downscaling techniques), which can then be expanded with additional synthetic generation as in Vormoor et al., 2017.





Historical daily measurements are available for the 2000-2014 period (MELCC, 2018). They include lakes inflows, levels and reservoir releases, and river discharges from the tributary and at the basin outlet.

Streamflow scenarios are made available by the CEHQ through the Quebec Water Atlas 2015 (CEHQ, 2015, MELCC, 2018). These hydrologic projections are based on climatic projections from the Natural Resources Canada data base of GCM simulations (CMIP5, Hydro-climatic Atlas, 2015) that were downscaled by the CEHQ.

A set of 501 time series was made available, spanning 30 years of daily inflows. The set contains 135 scenarios for a 1971-2000 reference period; and 366 scenarios for the 2041-2070 period. The 366 scenarios are based on 122 GCM projections,
from which 3 different downscaling techniques were applied: without bias correction, with quantile mapping or with delta quantile mapping (based on Mpelasoka and Chiew, 2009). In order to obtain the largest degree of variability, and find as many failure configurations as possible, all 501 time series are used indistinctively as input for the synthetic time series generation. The synthetic generator is the Kirsh-Nowak method (Nowak et al., 2010, Kirsh et al., 2013), made available online as Matlab® code by Quinn et al. (2017b), employed e.g. in Quinn et al., 2017a.

In order to expand the sample of the exposure space and explore less favorable conditions, the perturbation of synthetic inflows is performed by either modifying only the flow average without affecting the dispersion, or by affecting both. To increase the range of tested inflow volumes, a single change factor is applied in the first case, arbitrarily increasing all flow values at every time step by 50%. In the second case (perturbed dispersion) a varying factor multiplies flow values depending on their rank in the series distribution (factor 1 for the lowest, factor 1.5 for the highest flow). There are then 4 categories
of perturbation: volume only, dispersion, volume and dispersion, and none. Moreover each synthetic generation is performed twice for each available time series. We then get $501 \times 4 \times 2 = 4008$ synthetic time series, each containing 30 years of daily river discharges.

Similarly to other stress-test studies that generate inflow instead of climate time series (Feng et al., 2017), the selected driving variables (axes x and y of the response function) are the total annual inflow volume and a measure of the intra-annual
variability of streamflow. The intra-annual variability is here measured with the dispersion coefficient G, a measure also known as Gini coefficient in economics but employed too in hydrology (Masaki et al., 2014). It is similar to the variation coefficient used in other studies but bound between 0 and 1, which offers convenient interpretation: at G=0 all daily discharges in a year are equal, if G=1 the entire yearly run-off happens in a single day. Like the variation coefficient it allows for a second variable statistically independent of the total annual run-off volume. Here $q_i$ are the ordered daily discharges of a given year, N=365
days.

$$G = \frac{1}{N} \left( N + 1 - 2 \frac{\sum_{i=1}^{N} (N + 1 - i) q_i}{\sum_{i=1}^{N} q_i} \right) \tag{12}$$

### 3.3 Simulation and response surface

The model is built with HEC-ResSim, the Reservoir System Simulation software developed by the US Army Corps of Engineers (Klipsch and Hurst, 2007). It relies on a network of elements representing the physical system (reservoirs, junctions,





routing reaches), as well as the sets of operating rules. HEC-ResSim replicates the decision-making process applied to many
actual reservoirs through a rule-based modeling of operational constraints and targets.

Hydrologic inputs consist of 30 years long, daily river discharges for each sub-basin. The main outputs are daily water levels
in lakes, reservoir releases, as well as the discharge at the outlet. A complementary Jython routine is developed in order to
run HEC-ResSim in a loop to systematically load a large set of different hydro-climatic scenarios. Dam characteristics and

operational rules were provided by the Quebec Water Agency (MELCC, 2018).

The model is developed with a first set of operating rules (rule 1) expected to mimic the current operation of the system.
It reproduces measured daily releases over the 2000-2014 period. 4008 simulations are then run, each taking an input of
synthetic daily flow series spanning 30 years. In order to increase the density of the un-gridded exposure space sampling,
results are divided in 5 years periods. Such decomposition is deemed acceptable based on the reservoir system, which storage

capacity is designed for seasonal regulation, not multi-year, mitigating the effects of boundary conditions. It leads to a sample
of 24'048 points, each one representing a five-year simulation. Observation is independence not considered here, as the prime
objective is to maximize the diversity and noise of the sample.

Although the operating rules were designed taking into account all operating objectives, the present study focuses on the
flood control performance $p$. More specifically, it is the reliability (Hashimoto et al., 1982) of the system keeping the river

discharge at Weedon below 300 m$^3$s$^{-1}$. Mathematically, if $F(t)$ is the state of flooding at time step $t$, then $p$ is given by:

$$F(t) = \begin{cases} 0 \ \ if \ \ Q(t) \leq 300 \\ 1 \ \ if \ \ Q(t) > 300 \end{cases} \quad (13)$$

$$p = 1 - \frac{1}{T} \sum_{t=1}^{T} F(t) \quad (14)$$

The response function is built by representing $p$ as a function of the selected inflow characteristics (yearly volume and

dispersion). As developed in section 2, the separation of the exposure subspace is first performed through a performance
target $\theta$ set at $p \geq 0.95$. Alternatively the acceptability condition is defined by the fuzzy set within the bounds [0.93, 0.97[,
considering a 0.02 tolerance. Consequently, any given performance value $p$ has a membership degree of 0 for $p < 0.93$, and
equal to 1 for $p \geq 0.97$. Two different rules sets are tested: rule 1 that replicates the current management; and an instrumental
set rule 2 which slightly alters the anticipation and emergency release algorithm of the reservoirs. The current and alternative

sets of operational rules are compared based on the aggregated logistic regression, the analytical approximation and the error-
accounting membership function, or the relative position of the largest convex ranges of failure.

The LRCF method is also tried through an iterative sampling to evaluate its potential for computational parsimony. The
convex range of failure is thus first calculated on downscaled time series, including raw ones without bias correction (Fig. 7a).
Then each iteration expands the range by sampling pre-generated synthetic time series around its boundaries, and simulating

the water system with only those. Failure points constitute the new hull for the next iteration (Fig. 7b). Not finding failure
points is the exit condition. Results are compared to the LRCF calculation on the full sample.





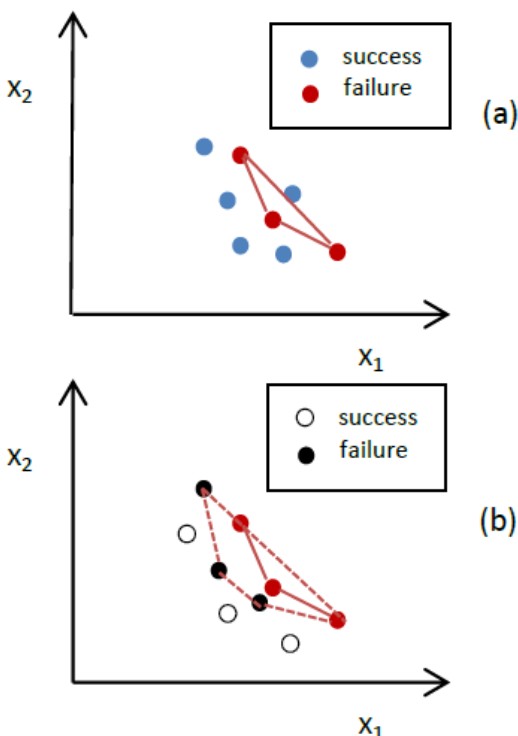

**Figure 7.** Concept for an iterative sampling of the LCRF. Simulations are first run for available hydr-climatic scenarios (a), then for successive random samples of synthetic time series (b).

## 4   Results

### 4.1   Simulations

The simulation is first run with 122 of the original time series made available by the CEHQ. These are the bias-corrected
rainfall / run-off simulations considered as the most reliable, corresponding to different radiative forcing scenarios. Taken by
5 year periods (thus 610 time series), all lead to flood control reliabilities superior to 0.97, above any considered performance
target. So both rule sets are considered successful in all these time series.

Simulations are then run for the much larger, and diverse, un-gridded sample of 4008 synthetic time series. The performance,
measured as the reliability of flood control, is evaluated for each 5-years period contained in the 4008 simulations of 30 years
(24'048 evaluations), for each of the two operation rules. The color scale represents the performance in Fig. 8 for each 5 year
time series. Describing variables are annual inflow volume at Lake Saint-François as x-axis and the dispersion of daily inflows
(or Gini coefficient) as y-axis. The large sample size and lack of gridded sampling reveal the noise of the response, although a
north-east / south-west anisotropy or gradient can be visually noticed.



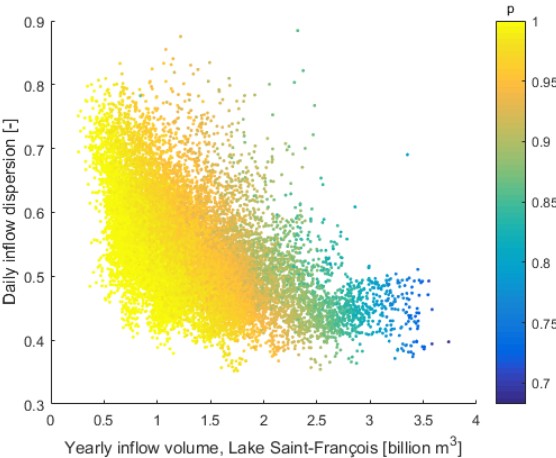

**Figure 8.** Response surface (rule 1). Performance p : flood control reliability

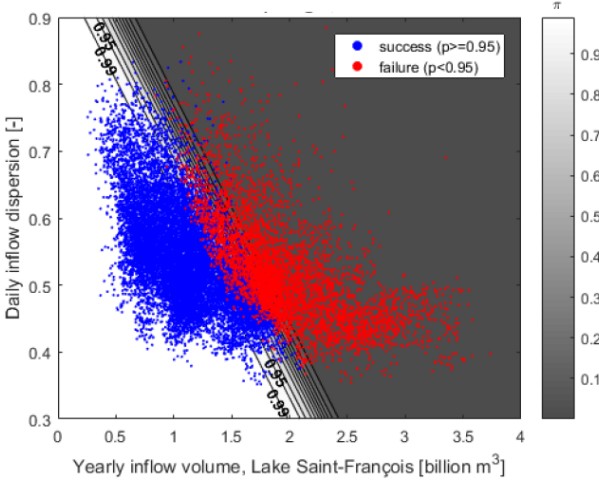

**Figure 9.** Logistic regression: probability of success $\pi$ for crisp target 0.95

## 4.2 Logistic surface aggregated over fuzzy target

The logistic regression is first performed with the response surface converted into crisp binary outcomes. Success is defined by p $\geq$ 0.95, failure by p $<$ 0.95. The logistic surface provides the probability of success $\pi$ at any coordinates (Fig. 9). Depending on the risk attitude of stakeholders or decision-maker, the surface can be divided in success and failure regions for specific probabilities of success ($\pi$-cuts, eq. 9), like 0.95 or 0.99 as shown on the figure. The approximation was done with the Matlabs® function *mnrfit*. The deviance of the fit – an expression of likelihood of the model, 0 being a perfect model – is

0.0085.





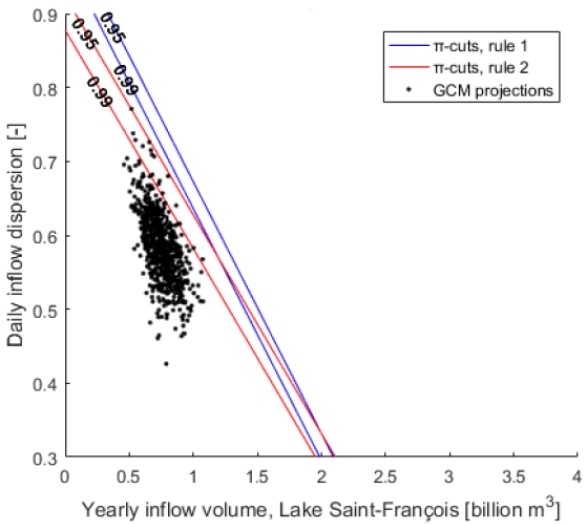

**Figure 10.** Compared logistic regressions, rules 1 and 2, vs GCM hydroclimatic scenarios. Crisp target : $p \geq 0.95$

As in Kim et al.(2019) probability cuts can be contrasted with the projection of downscaled time series from GCMs on the response surface (Fig. 10). A way to compare the two management rules, besides the relative position of their $\pi$-cuts, is the number of projections falling out the $\pi$-cuts. For rule 1, no GCM projection – taken as 5 year portions - falls out of the 0.99 $\pi$-cut, i.e. the space in which $\pi(p \geq 0.95) \geq 0.99$. For rule 2, 17 projections fall out of the 0.99 $\pi$-cut, one of them falling too out of the 0.95 $\pi$-cut.

While all these downscaled time series lead to successful performances, showing reliability values above 0.97 for any rule, their coordinates, thus their corresponding driving variables, can still fall outside of a $\pi$-cut. With rule 2, a scenario sharing the same properties $x_1$, $x_2$ - yearly inflow volume and daily inflow dispersion – with a successful GCM projection could still lead to failure.

The fuzzy performance target is then integrated to the calculation. Success is not anymore a set defined with the satisfaction of a crisp target of 0.95 for flood reliability, but with a fuzzy target between 0.93 and 0.97. Reliability values above 0.97 are considered full success, and below 0.93 full failure. In between, the membership function can be either linear or sigmoid. For both cases, and for both rule sets, the logistic regression is performed 10 times for 10 $\alpha$-cuts corresponding to a uniform sampling of $\alpha$-levels (see section 2.2.1.). The aggregated logistic regression at every coordinate is the average of the 10 logistic regressions, each one considering a single $\alpha$-cut as the crisp set over $p$ that defines successful outcomes. Figure 11 compares the crisp logistic regression and the aggregations over linear and sigmoid fuzzy targets. The averaging effect can be noted in the wider transition zone, which becomes steeper at its center when applying a sigmoid fuzzy target.

Both rule sets can be compared through their aggregated possibility ($\Pi$) cuts, e.g. at 0.95 and 0.99 (fig 12). With rule 1 applied, zero GCM projections fall outside of the 0.95 possibility cut, and 8 fall out of the 0.99 cut. With rule 2, 6 projections fall out of the 0.95 cut, and 46 out of the 0.99 cut.



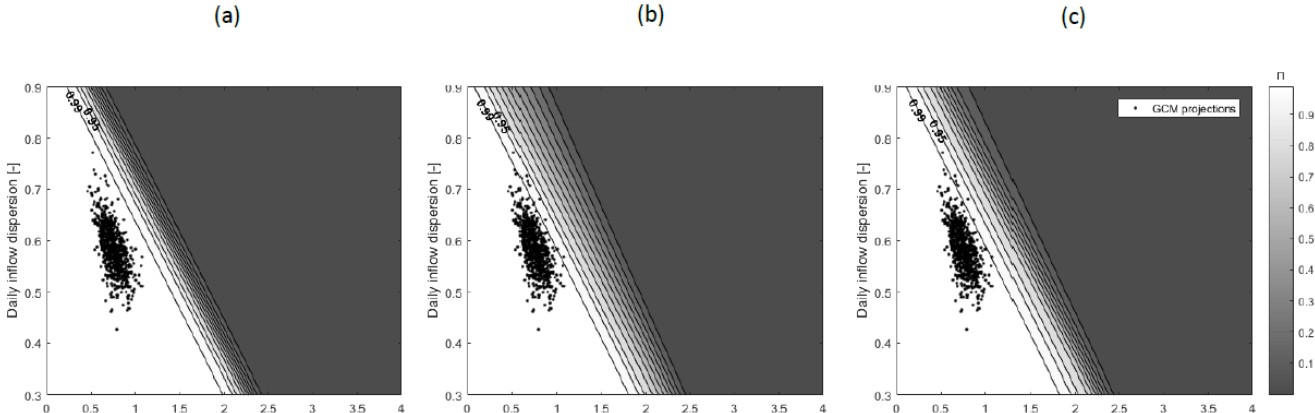

**Figure 11.** Compared logistic surfaces, rule 1 (a) 0.95 crisp target (b) [0.93 0.97] linear fuzzy target (c) [0.93 0.97] sigmoid fuzzy target

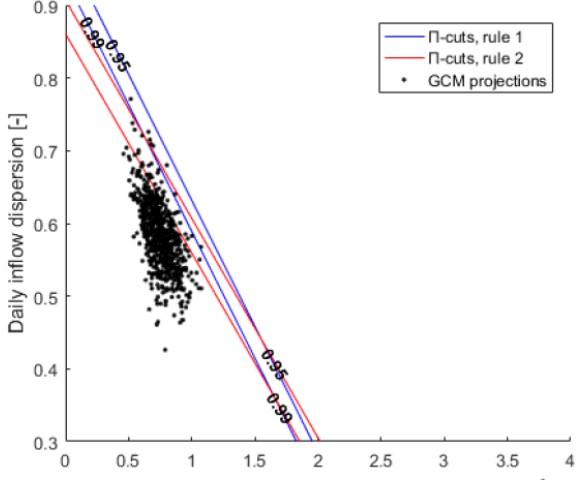

**Figure 12.** compared possibility cuts (Π-cuts, rules 1 and 2) vs GCM hydro-climatic projections, sigmoid fuzzy target

### 4.3 Bivariate surface approximation

The second method consists in computing analytical functions, one for each rule, to fit the available sample, in this case with a bivariate quadratic approximation (Fig. 13). The resulting error R = 0.03 (selected here as the 95% quantile in the error distribution) is used to modify the membership function $\mu$ of the success fuzzy set $A_\mu$, with a moving average with 2R-sized window (section 2.2.2.).

With an explicit, deterministic function and a modified membership function $\mu_R$, the membership degree to the fuzzy set $A_{\mu R}$ allows to map success (white, membership degree 1) and failure (black, membership degree 0) sub-spaces with a continuous transition (Fig. 14a).




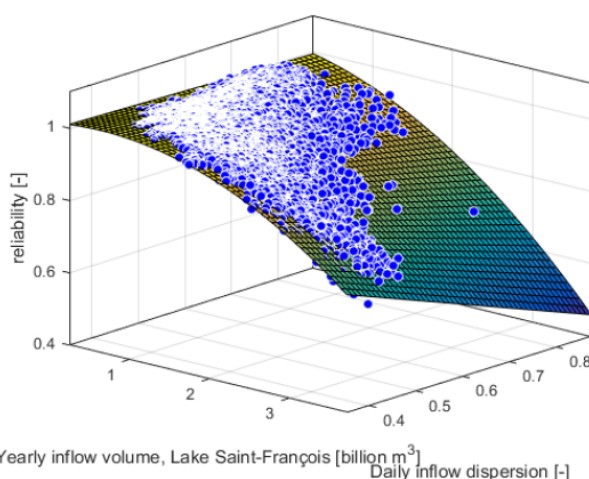

**Figure 13.** quadratic approximation of the response surface.

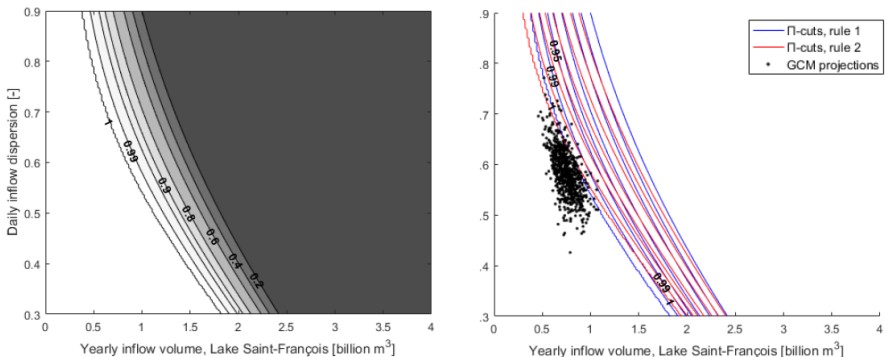

**Figure 14.** Possibilistic surface, quadratic approximation (a) Π-cuts, rule 1 (b) compared Π-cuts vs GCM hydro-climatic projections

GCM time series, again, while by themselves showing fully accepted performances (membership degree equal to 1), are
partially located in the $9^{th}$ decile of the response surface.

Comparison between operation rules seems however less conclusive in this case (Fig. 14b), at least around the GCM-based
projections. No projection falls out of the 0.95 Π-cut. 4 projections fail to meet the 0.99 Π-cut for rule 1, 6 for rule 2 (rule 1 thus
being slightly better). For rule 1, 44 projections do not reach an aggregated membership of 1, 40 for rule 2, swapping positions
in this case. It can be noted that, while both rules are similar in the vicinity of GCM-based projections, rule 1 performs better in
the low inflow, high dispersion zone, and rule 2 in the high inflow, low dispersion. The method allows however for a non-linear
relation between $x_1, x_2$ variables and performance, as opposed to the logistic regression. Dispersion has a varying effect on
performance depending on total inflow, with an increasing slope as yearly inflow grows.





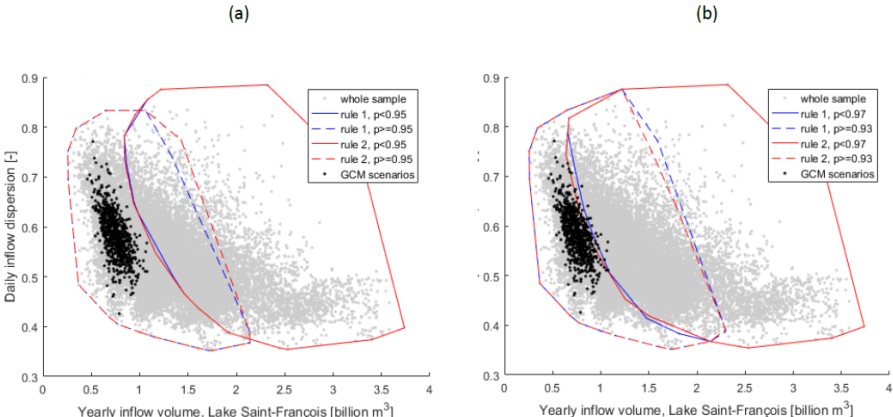

**Figure 15.** Compared largest convex ranges of success (LCRS) and failure (LCRF) vs GCM hydro-climatic projections (a) for crisp target 0.95 (b) for fuzzy target [0.93 0.97].

## 4.4 Largest convex range of failure (LCRF)

The third approach to identify accepted and rejected sub-spaces for each management rule is the comparison of respective
convex hulls, each hull representing a possibility sub-space of success/failure of the system respective to flood reliability. It heavily relies on outliers and thus represents an upper possibility bound, here called the maximum convex range of success or failure (LCRF). It answers the question "where can the system possibly fail or succeed" given a sample of points. Figure 15a shows the maximum range of success (dashed line) for both rule 1 (blue) and rule 2 (red), that includes all values with flood reliability superior or equal to the acceptability threshold, 0.95, here considered as a crisp value. The solid line hulls represent
the maximum range of failure, containing all values with flood reliability inferior to 0.95. The overlap between maximum ranges is a transition zone. With a similar range of failure and a larger range of success, the rule set 2 (red) would this time be considered as superior to the rule set 1.

In Fig. 15b, sub-spaces are defined with a fuzzy performance target, with a +/- 0.02 tolerance and bounds defined as [0.93, 0.97[ in section 2.1.2. The largest range of success now accepts candidate values that are partially accepted (partial successes,
$p \geq 0.93$), so it considers success as the largest $\alpha$-cut of $A_\mu$. The shape of the membership function has no influence here. Respectively the largest convex range of failure considers as success the smallest $\alpha$-cut of $A_\mu$, it now accepts candidate values considered as partial failures of the system (p<0.97).

Figure 15 also highlights (black dots) the position of the downscaled, bias-corrected hydro-climatic scenarios based on GCM projections. Again in this case, even if none of such time series is considered as even partial failure (flood reliability p>0.97
for all of them), with a fuzzy definition of target performance some of these time series can fall within a maximum range of failure.

Figure 15b shows a varying relative performance of both rules depending on the location in the exposure space. Rule 1 has a larger maximum range of success. Maximum ranges of failure for both rule sets sometimes switch their relative position: rule 2





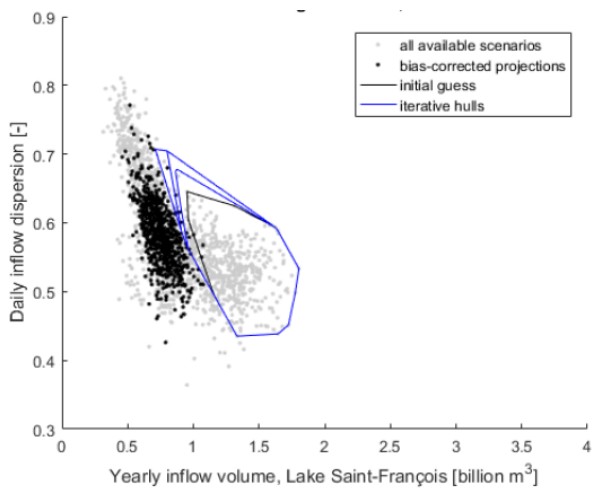

**Figure 16.** Iterative sampling of the LCRF, rule 1.

usually performs worse, but not systematically. Here again the GCM projections allow for an additional weighting, pointing at
the priority region to analyze. In this case rule 1 shows a better performance in the vicinity of climate projections. 60 segments
of climate projections out of 610 (9.8%) fall within the maximum space of failure with rule 2 while it is only the case for 12 of
them with rule 1 (2%).

Given that a limited number of outliers define the maximum ranges, an ex-post sampling algorithm is tested to see if the
number of simulations can be reduced and reach similar results. Only the range of failure is tried here. Starting from the
original 3006 time series from 501 available run-off simulations (including those based on raw downscaled rainfall models
without bias correction), the convex range of failure is calculated then "stretched" by iterative sampling. At each iteration, a
new random sample of 1000 synthetic time series is selected within a normalized radius of 0.5 from both the previous hull and
the bias-corrected scenarios. New failure points are added to the previous set of failures and constitute the new range of failure,
other points are discarded. The number of bias-corrected scenarios included in the final hull is an evaluation of the quality of
the algorithm; the closer to the result obtained with the full sample of synthetic time series the better.

Figure 16 shows an example of how the range of failure grows at each iteration with rule 1. Three iterations are enough in
this case to reach 11 GCM scenarios within the range of failure. It is the same number as obtained with the full sample (Fig.
15b) so it is the best possible result for the algorithm. The number of required 5-year simulations of the water system is 6006,
3006 for the first guess plus 3000 from the iterations, compared to the 24048 from the full synthetic sample. Figure 17 shows
a sensitivity analysis with 100 runs for both rules. While a majority of runs reach a number of GCM scenarios within failure
range that is close to the full sample (10-11 for rule 1, 59-60 for rule 2), a considerable number of runs falls short, although
still far from switching performance between the two rules. Numbers on top of columns show the range of required iterations,
between 3 and 9 overall. The iterative process can thus divide the overall simulation time by 2 to 4.





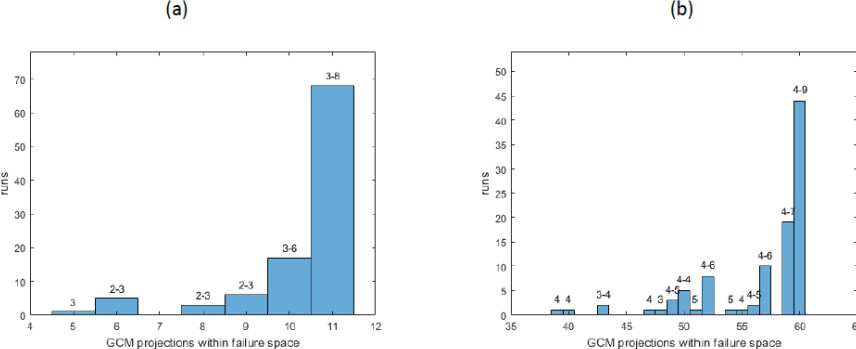

**Figure 17.** Sensitivity test of the iterative LCRF: normalized search radius 0.05, 1000 simulations by iteration. Numbers on columns give the range of required iterations. (a) rule 1 (b) rule 2.

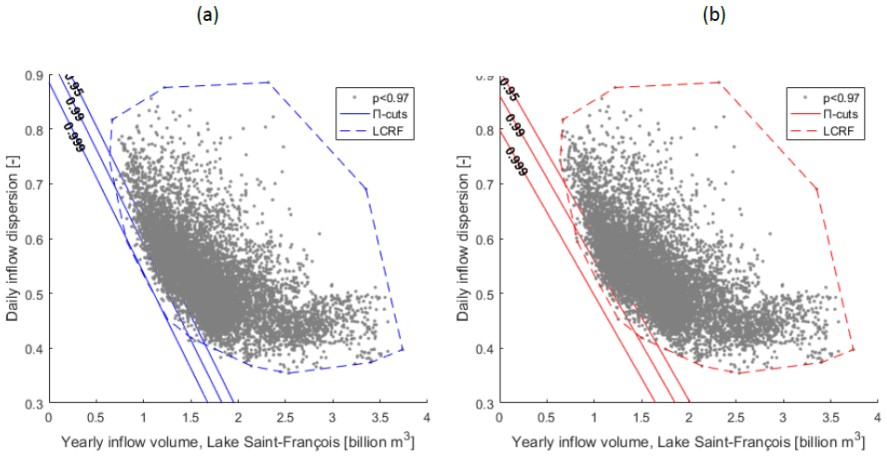

**Figure 18.** Logistic possibility cuts, sigmoid fuzzy target vs LCRF. (a) rule 1 (b) rule 2.

Finally, Fig. 18 combines the largest convex ranges of failure with the fuzzy logistic surfaces. As the logistic regression is
itself a sigmoid approximation that cannot take values of 0 or 1, the LCRF can be a complement to remind the actual position
of the point of failure that is the farthest on the gradient line. An empirical distribution projected on the gradient line of the
logistic regression would reach a value of 1 at that point.

## 5  Discussion

By itself, the stress-test approach is a departure from a probabilistic framework towards a possibilistic one. It asks what
situations lead to a system failure, instead of evaluating the system for the most probable future. Since response surfaces are
not deterministic, further information of irreducible uncertainty must be incorporated through e.g. the use of logistic regression


(Kim et al., 2019). In this paper, we further consider that the threshold employed to define success might be itself ambiguous or contentious. The fuzzy or possibilistic framework (Zadeh 1965, 1978; Dubois and Prade, 1988), often used in decision-making analysis provides the analytical tools to incorporate an uncertainty that is not probabilistic in nature, the ambiguity of a decision
target, within the popular stress-test tool that itself seeks to depart from probabilistic approaches.

Applying a fuzzy target would be straightforward for a deterministic response surface, each performance value on the exposure space being mapped to a degree of success between 0 and 1. This study explores how to combine a fuzzy definition of success or failure with the remaining hydro-climatic uncertainty of the response surface, and compares different methods and interpretations.

As a first option, the aggregated logistic regression measures within a single possibility value the probabilistic information of the regressions and the fuzzy definition of the performance target. The shape of the membership function also affects how the $\alpha$-cuts are sampled, thus allowing for different interpretations of ambiguity and decision theories. A linear membership function translates a form of neutrality towards marginal gains or losses within the fuzzy boundary. A sigmoid shape gives more weight to the median $\alpha$-cut , corresponding to a degree of success of 0.5, and diminishing marginal improvement or
loss the further the $\alpha$-cut is from the median, which can be thought as based on prospect theory (Kahneman and Tversky, 1979). The relative performance of the compared rules, however, is not here altered by the inclusion of the fuzzy target, so the resulting decision is not affected. It still might be the case when the response surfaces have different slopes and gradient directions depending on the tested alternative.

The analytic approximation by a quadratic function similarly proposes a continuous measure of possibility over the exposure
space, but has the advantage of identifying non-linearity between describing variables, which the logistic regression cannot. Possible drawback are that, while the logistic regression considers equally all results that fall out of the fuzzy target, extreme performance values here shape the fitted surface and might have an influence that they do not have in the decision process. And of course, the sample might be unadapted to any fitting attempt. As for comparing options, the method remains less conclusive, in the vicinity of GCM projections both rules cannot be easily sorted out. Importantly, it confirms however the possibility of
diverging slopes and directions, as the preference between the two rules can switch depending on the position in the exposure space, and therefore that considering fuzzy targets could very well alter the preference between rules. Varying preference depending on the exposure space is also a case for adaptive management.

The largest convex ranges of success or failure provide an upper possibility bound and thus easily integrate the fuzzy target by either maximizing or minimizing the $\alpha$-cut of the fuzzy set of success. The shape of the membership function has no impact
here, only its bounds. An advantage of the method is its consistency: if the whole philosophy of a stress test is asking where the system can *possibly* fail, then it is good to look and to prepare for the least probable cases of failure, those in a region where success is *almost* guaranteed.

However, the convex hulls are obviously highly reliant on the generation of synthetic scenarios, which has its intrinsic randomness. The largest convex range of failure focuses on a very limited number of vertices as hydro-climatic situations of
interest, the least probable, but still possible, configurations of failure; but this limited set also entails a strong sensitivity to a specific realization, with a specific generator. More generally, the impact of the choice of synthetic generator is a growing





concern (Nazemi et al., 2020) and should be further studied, possibly integrating the potential differences within a possibilistic approach.

The fact that they are defined by a very small number of points could allow for much shorter simulation times with the right
sampling method. However a first trial with a simple search algorithm shows for now an important spread in results for limited computational gains. Convex hulls also remain a straightforward tool in spatial or point process analysis and, like the logistic regression, are not always suited for non-linear relations between describing variables. Refinements would be needed, possibly at the cost of more degrees of freedom or assumptions if more parameters are needed, like in alpha-shapes or more advanced clustering tools. The time series that constitute the vertices of the hulls can be further perturbed, to see if the range of success
or failure can be "stretched" and look for a *physical* possibility (where is it physically possible for the system to succeed or fail).

Un-gridded and un-aggregated sampling here allows exploring more comprehensively the variability of the response, which can be more consistent with the whole stress test approach for certain systems. It also makes use of existing streamflow scenarios, but it has drawbacks. One advantage of usual stress-tests is their scenario-neutral property (Prudhomme et al., 2010):
the extensive computation of the response surface only needs to be done once, further information on future conditions can be projected directly on it. Here stream-flow scenarios, based on GCM projections, were used to generate synthetic time series in order to obtain the granular, non-gridded response. This creates a link between downscaled scenarios and the response that is usually avoided. The underlying assumption is that between the number of scenarios, the different types of bias correction (or lack of thereof) and the imposed perturbations, the diversity of the synthetic time series leads to a relative independence from
the initial bias of GCM simulations. Besides, the generation of synthetic time series always relies on available data on one way or another; a "scenario-neutral" generation could rely on historical observations and be skewed towards a conservative bias in face of for example, brutal climate shifts. The present generation method prioritizes sample diversity, but its assumptions could be further examined, such as the lack of independence between observations and thus the applicability of the logistic regression. Likewise, the choice of describing variables was not the focus of the study but should be subject to an initial comparison of
predicting values within a larger number of predictors.

Another trade-off from such synthetic generation is the uneven sampling, denser at the vicinity of available streamflow scenarios from downscaled rainfall series. The tool to generate un-gridded sampling should ideally ensure a balanced sample density over the response surface. This study used like others the streamflow scenarios from GCM projections as a prioritization tool and focuses on their vicinity, thus paying less attention to the sampling density in other areas.
The integration of uncertainty and ambiguity quantification within the response surface tool could allow for further aggregation options in a multi-objective problem (like in Poff et al., 2016, Kim et al., 2019), while easily controlling its two separate components, response uncertainty and target ambiguity. Other sources of uncertainty could also be added and combined, like ambiguity about the streamflow threshold that defines a state of flooding, the goodness of fit for the approximations, or the expert judgement or trust on data quality.





# 6 Conclusions

We explore in this study how to integrate fuzzy performance targets within uncertain response surfaces in decision-centric vulnerability assessments. Three methods are proposed to produce a possibilistic surface. Aggregating logistic regressions over $\alpha$-cuts combines probability of success and target ambiguity in a single measure. Using a quadratic approximation of the response surface itself allows for non-linear relations. The largest convex ranges seek upper bounds for the possibility of success or failure. Two possible management rules are compared for the Upper Saint-François reservoir system in Canada. Aggregated logistic regression and largest range of failure show complementary ways to integrate fuzzy targets and differentiate failure domains, with respective advantage and limitations. For continuous approximations, fit quality could be integrated in the final uncertainty measure. The largest convex range method can be refined by further perturbation of the streamflow series on the vertices, in order to find a physical boundary to success and failure.

Challenging old probabilistic assumptions, notably in a climate crisis context, brings new tools that also imply new choices and degrees of arbitrariness. How to transparently elaborate fuzzy targets jointly with stakeholders, or the choice of a synthetic scenario generator, are necessary research continuations. The presented approach enables further work on multi-objective problems and aggregation choices. The framework here introduced to solve a practical challenge can be consolidated from a more theoretical perspective, from both possibility theory and decision making under deep uncertainty.

*Code and data availability.* The data can be provided upon authorization from the MELCC, Québec, Canada (Ministère de l'Environnement et de la Lutte contre les Changements Climatiques). The codes required to reproduce the results are available upon request (thibaut.lachaut1@ulaval.ca).

*Author contributions.* TL and AT conceptualized the study. TL developed the methods, models and simulations, and drafted the manuscript. AT acquired the funding and provided extensive supervision.

*Competing interests.* The authors declare that they have no conflict of interest.

*Acknowledgements.* The work was supported by a project from Ministère de l'Environnement et de la Lutte contre les Changements Climatiques (MELCC, Québec, Canada) entitled "Étude visant l'adaptation de la gestion des barrages du système hydrique du Haut-Saint-François à l'impact des changements climatiques dans le cadre du Plan d'action 2013-2020 sur les changements climatiques (PACC 2020)". This study does not represent the views of MELCC. The authors would like to thank Louis-Guillaume Fortin, Richard Turcotte and Julie Lafleur from the CEHQ for the fruitful discussions and the knowledge of the reservoir system, Alexandre Mercille and Xavier Faucher who contributed to earlier versions of the HEC-ResSim model, and Jean-Philippe Marceau who developed the iterative Jython routine.



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
