# Peer review of "Possibilistic response surfaces combining fuzzy targets and hydro-climatic uncertainty in flood vulnerability assessment"

_Hydrology and Earth System Sciences, 2020_

## Referee Comment (RC1) · Anonymous Referee #1 · 15 Jul 2020

Possibilistic response surfaces combining fuzzy targets and hydro-climatic uncertainty in flood vulnerability assessment

This paper develops new approaches for bottom-up decision making approaches considering joint uncertainties in the system response surface and the performance target. Three methods are proposed: a fuzzy logistic regression, an analytical approximation, and a convex hull method. A case study of flood risk in Canada is used to illustrate the methods.

[Figure]

The paper identifies important challenges in bottom-up methodology, and the proposed methods are new to the field while also drawing on historical developments in decision theory. However, the results do not clearly illustrate the benefits of the new approaches, and may introduce more complexity. I believe this can be resolved with substantial revisions, as the authors have done a nice job with the motivation and methods description.

1. My first concern is how the methods treat hydroclimatic uncertainty in the response surface. The paper notes that the variables sampled in the response surface only partially cover the space of possible uncertainties, which I agree with. However, I would not say that this can be captured by the uncertainty in the fit of the response surface using logistic regression. The uncertainties we are most concerned with are the hydroclimate timeseries and natural variability, which will not be captured using this approach.

It is not reasonable to expect the authors to find a way to quantify this uncertainty, which would be a different study altogether. But the claims about the types of uncertainties considered should be aligned with the experiment.

2. The results section is quite long, and does not clearly show the value of the new approaches within the decision-making context. The paper would be much stronger if the authors could resolve this. I would suggest refining and shortening the figure sequence to more clearly show the differences between the standard response surface and the new methods, especially if there is a way to highlight differences in the decisions that would result.

At present, the results seem to show that the new approaches yield only small differences from the standard stress-test, which may not be significant in the context of other uncertainties in hydroclimate as mentioned above.

3. The methods proposed by the authors provide a more formal way to incorporate uncertainties not usually considered in bottom-up modeling studies. However I am not sure of its practical value, because it replaces the subjective choice of a single threshold with the choice of a membership function, which is perhaps even more difficult to define. The authors recognize this challenge in the conclusion. This limitation would be somewhat resolved if the results clearly showed an advantage to the more complex uncertainty representation.

Minor points - The introduction starts very broad, and could be edited for clarity - The bibliography contains references not cited in the paper, and vice versa

---

## Referee Comment (RC2) · Anonymous Referee #2 · 9 Aug 2020

Lachaut and Tilmant introduce the concept of "possibilistic surfaces" to describe conditions under which success or failure of a water resources system is possible, where regions of "possibility" are defined in three different ways: 1) using logistic regression and defining success regions as conditions under which the logistic regression predicts success in meeting a threshold of satisfaction with at least some probability p, 2) using fuzzy performance thresholds in which a hard success/failure threshold does not need to be defined for a logistic regression model, rather a fuzzy membership function is used to assign continuous performance values to fuzzy sets, and 3) using convex

hulls to define regions of success based on the outer bound of scenarios in which performance was found to be acceptable. The authors also discuss benefits of employing non-gridded sampling of conditions under which to evaluate water system performance to generate these surfaces.

Of the 3 possibilistic surfaces introduced by the authors, I believe only the second is new to the literature. As noted by the authors, Kim et al. (2019) use logistic regression to define success and failure regions. However, the authors do not discuss Quinn et al. (2018), who used logistic regression as described in this paper to define success/failure regions that account for stakeholders' different levels of risk aversion by choosing different probabilities of success from the logistic regression to define the boundary. The authors also state that logistic regression cannot capture nonlinear relationships in the mapping of climate conditions to success/failure, but this is not true. One can easily incorporate interaction or nonlinear predictors in a logistic regression just as in a linear regression. See Hadjimichael et al. (2020) for an example. Other studies which use logistic regression for scenario discovery that were not cited by the authors include Lamontagne et al. (2019) and Marcos-Garcia et al. (2020).

With respect to the convex hull representation of possibilistic surfaces, this sounds like info-gap decision theory (Ben-Haim, 2006), which the authors do not discuss in the paper. It is not clear what their method contributes beyond this approach. It is also worth noting potential problems with this approach. One, which is briefly described by the authors, is that if the failure boundary is not convex, it could be too conservative. For example, a failure region like the red region in the attached figure could be estimated by logistic or linear regression with an interaction term between the two factors on each axis to capture the non-convexity. The convex hull, however, would include everything to the right of the black line, which includes a substantial region of successes in blue. But a convex hull might not always be more conservative like the authors imply. This is because it is defined by the realized values from their model simulations. As discussed by the authors with respect to their logistic regression model, none of the GCMs met

their failure definition, but the probability of success in those worlds did not always meet their threshold of acceptability. They might not fall within the convex hull of failures, though, making the convex hull a less conservative definition in that case.

Where I think the authors have introduced a new approach to the literature is in combining logistic or linear regression with fuzzy set membership. The question is, what value does this method add to the alternative approaches? I think this should be the focus of the paper, and it is not currently clear what that value is. Personally, I find the first and third approaches more intuitive. It is easy to understand what a probability of success represents, so defining success regions based on probability contours from a logistic regression makes sense to me. Similarly, it is easy to understand a failure region defined by lines connecting the farthest scenarios in which failures have occurred. I find fuzzy sets much harder to interpret, and more subjective to define. But I think it could provide value in that no hard success/failure threshold has to be assumed if using it with linear regression, whereas this is not true for the other two approaches. It would be helpful to expound more on this benefit, and the differences that come out of using this approach as opposed to Method 1. It is likely no method dominates all others, but why is this new method on the Pareto front of options? This needs to be better emphasized by comparing and contrasting the regions that come out of the alternative approaches.

Finally, the authors discuss shortcomings of using gridded scenarios to build models of success/failure regions, but they never compare their non-gridded sampling to a gridded sample to illustrate its claimed superiority. I suggest the authors remove this argument entirely as it is a secondary argument anyway, and is never actually illustrated. Please see the annotated manuscript for additional, more minor comments.

* * *
[Figure]

**Fig. 1.**

**Supplement:**

[revised manuscript text omitted]

---

## Author Comment (AC1) · 24 Sep 2020

We would like to thank the referee for his/her constructive comments.

*"This paper develops new approaches for bottom-up decision making approaches considering joint uncertainties in the system response surface and the performance target. Three methods are proposed: a fuzzy logistic regression, an analytical approximation, and a convex hull method. A case study of flood risk in Canada is used to illustrate the*

*methods.*

*The paper identifies important challenges in bottom-up methodology, and the proposed methods are new to the field while also drawing on historical developments in decision theory. However, the results do not clearly illustrate the benefits of the new approaches, and may introduce more complexity. I believe this can be resolved with substantial revisions, as the authors have done a nice job with the motivation and methods description.*

*1. My first concern is how the methods treat hydroclimatic uncertainty in the response surface. The paper notes that the variables sampled in the response surface only partially cover the space of possible uncertainties, which I agree with. However, I would not say that this can be captured by the uncertainty in the fit of the response surface using logistic regression. The uncertainties we are most concerned with are the hydroclimate timeseries and natural variability, which will not be captured using this approach.*

*It is not reasonable to expect the authors to find a way to quantify this uncertainty, which would be a different study altogether. But the claims about the types of uncertainties considered should be aligned with the experiment."*

We agree with you that the hydroclimatic uncertainty cannot be captured by the uncertainty in the fit of the response surface. In the revised manuscript we will better explain that the main objective is to explore how we can integrate fuzzy thresholds in vulnerability assessment approaches, and how we can combine this ambiguity with the uncertainty inherent to a bivariate response. The first method indeed uses the logistic regression previously employed (Kim et al. 2019) as one of the ways to convey this uncertainty, proposing a division of the exposure space by probability of success. This probability of success at each coordinate aims at capturing part of the hydro-climatic uncertainty that the 2 variables of the exposure space do not capture. We do not here consider the additional uncertainty on the fit of the logistic regression itself, though we mention it could be incorporated.

*"2. The results section is quite long, and does not clearly show the value of the new approaches within the decision-making context. The paper would be much stronger if the authors could resolve this. I would suggest refining and shortening the figure sequence to more clearly show the differences between the standard response surface and the new methods, especially if there is a way to highlight differences in the decisions that would result.*

*At present, the results seem to show that the new approaches yield only small differences from the standard stress-test, which may not be significant in the context of other uncertainties in hydroclimate as mentioned above."*

Thank you for your suggestions. Based on both referee comments we realize that the main objective of the paper, which is on the consideration of fuzzy thresholds in vulnerability assessment approaches, did not come across clearly. The confusion comes from the fact that this incorporation was analyzed for three alternative methods to generate regions of success and failure. In the revised version, we will shorten the material and method section by focusing on fuzzy thresholds combined with one generating method: the logistic regression. This choice is motivated by the fact that this approach has received a lot of attention recently in the literature (in addition to Kim et al., 2019; the paper will cite Quinn et al., 2018, Lamontagne et al., 2019, Hadjimichael et al, 2020; Marcos-Garcia et al., 2020).

We acknowledge the interest of discussing the effect on outcomes. However, when comparing the effects of using a crisp and fuzzy threshold, the crisp threshold is only counterfactual, not an alternative option to be compared to. It helps to visualize how fuzzy thresholds affect the division of the exposure space in regions, but we assume that the crisp threshold is not available in the first place. This should be further emphasized in the result section.

Cases where a fuzzy threshold can lead to a different decision – again, compared to a counterfactual crisp threshold – are also worth discussing. In the attached figure

(to be added in section 2 of the revised manuscript), we see that it is theoretically the case if the response functions of two options have different slopes. This can also be illustrated with other case studies, e.g. Quinn et al. (2017) where an improvement in moderate flooding can make extreme floods worse, thus possibly changing the slope of performance as function of stressors. However, in the present paper, the difference in slopes between the alternative options is small and does not lead to a different decision in this case.

Still, we think that this case-study specific result does not diminish the validity of the underlying research question: how do we handle non-clearly defined (fuzzy) thresholds in bottom-up, vulnerability assessment studies? The paper needs to make clear that the method is not an alternative but an extension of bottom-up vulnerability assessment studies for particular situations, whereby crisp thresholds do not exist due to a variety of reasons including the lack of consensus amongst stakeholders, the ambiguous definition of the associated objective, etc.

*"3. The methods proposed by the authors provide a more formal way to incorporate uncertainties not usually considered in bottom-up modeling studies. However I am not sure of its practical value, because it replaces the subjective choice of a single threshold with the choice of a membership function, which is perhaps even more difficult to define. The authors recognize this challenge in the conclusion. This limitation would be somewhat resolved if the results clearly showed an advantage to the more complex uncertainty representation."*

This is an interesting point. Substituting a crisp with a fuzzy threshold indeed requires the definition of a membership function, which is not necessarily straightforward. But this issue is well known in fuzzy set theory and has been extensively investigated by various authors in several application fields, including multicriteria analyses (see Bouchon-Meunier et al., 1996; Haber et al., 2002; Garibaldi et al., 2003; Wu, 2012;

[Figure]

Sadollah, 2018). Our position is that this difficulty should not preclude the development/refinement of bottom-up approaches so that they can handle non-crisp thresholds. Ultimately, it is up to the analyst and decision maker to decide whether the incorporation of a fuzzy threshold is worth the additional effort. In the revised version, we will discuss this issue in the concluding remarks and suggest further readings on how to interactively select membership functions.

*"Minor points - The introduction starts very broad, and could be edited for clarity - The bibliography contains references not cited in the paper, and vice versa"*

Thank you for highlighting this, the introduction will be revised with a more focused start. The errors in the bibliography are fixed.

References:

Hadjimichael, A., Quinn, J., Wilson, E., Reed, P., Basdekas, L., Yates, D., Garrison, M. (2020) Defining robustness, vulnerabilities, and consequential scenarios for diverse stakeholder interests in institutionally complex river basins. Earth's Future, e2020EF001503.

Kim, D., Chun, J. A., Choi, S. J. (2019). Incorporating the logistic regression into a decision-centric assessment of climate change impacts on a complex river system. Hydrology Earth System Sciences, 23(2).

Lamontagne, J. R., Reed, P. M., Marangoni, G., Keller, K., Garner, G. G. (2019). Robust abatement pathways to tolerable climate futures require immediate global action. Nature Climate Change, 9(4), 290-294.

Marcos-Garcia, P., Brown, C., Pulido-Velazquez, M. (2020). Development of climate impact response functions for highly regulated water resource systems. Journal of Hydrology, 125251.

Quinn, J. D., Reed, P. M., Giuliani, M., and Castelletti, A. (2017). Rival framings: A framework for discovering how problem formulation uncertainties shape risk management trade-offs in water resources systems, Water Resources Research, 53, 7208–7233.

Quinn, J. D., Reed, P. M., Giuliani, M., Castelletti, A., Oyler, J. W., Nicholas, R. E. (2018). Exploring how changing monsoonal dynamics and human pressures challenge multireservoir management for flood protection, hydropower production, and agricultural water supply. Water Resources Research, 54(7), 4638-4662.

R. E. Haber, R. Haber, A. Alique, and S. Ros (2002). Application of knowledge-based systems for supervision and control of machining processes" Handbook of software engineering and knowledge engineering, vol. 2, pp. 673-710.

J. M. Garibaldi and R. I. John (2003). Choosing membership functions of linguistic terms. The 12th IEEE International Conference on Fuzzy Systems, 2003. FUZZ '03., St Louis, MO, USA, 2003, pp. 578-583 vol.1, doi: 10.1109/FUZZ.2003.1209428

B. Bouchon-Meunier, M. Dotoli, B. Maione and D. Bari. (1996). On The Choice Of Membership Functions In A Mamdani-Type Fuzzy Controller.

Wu D. (2012). Twelve considerations in choosing between Gaussian and trapezoidal membership functions in interval type-2 fuzzy logic controllers. IEEE International Conference on Fuzzy Systems (FUZZ-IEEE); Brisbane, QLD, Australia

Sadollah, A.(2018). Introductory Chapter: Which Membership Function is Appropriate in Fuzzy System? 10.5772/intechopen.79552.
* * *
[Figure]

**Fig. 1.** With a crisp threshold $\theta$, rule 2 has a larger success region A2.

Performance r

Rule 1

Rule 2

Θ1

Θ2

stressor X

**Fig. 2.** With a fuzzy threshold ($\theta 1$, $\theta 2$), Rule 2 has a larger "at least partial" success domain S2, but a smaller "full" success domain C2, than Rule 1.

---

## Author Comment (AC2) · 24 Sep 2020

We would like to thank the referee for this thorough review.

*"Lachaut and Tilmant introduce the concept of "possibilistic surfaces" to describe conditions under which success or failure of a water resources system is possible, where regions of "possibility" are defined in three different ways: 1) using logistic regression and defining success regions as conditions under which the logistic regression predicts*

*success in meeting a threshold of satisfaction with at least some probability p, 2) us-ing fuzzy performance thresholds in which a hard success/failure threshold does not need to be defined for a logistic regression model, rather a fuzzy membership function is used to assign continuous performance values to fuzzy sets, and 3) using convex hulls to define regions of success based on the outer bound of scenarios in which per-formance was found to be acceptable. The authors also discuss benefits of employing non-gridded sampling of conditions under which to evaluate water system performance to generate these surfaces.*

*Of the 3 possibilistic surfaces introduced by the authors, I believe only the second is new to the literature. As noted by the authors, Kim et al. (2019) use logistic regression to define success and failure regions. "*

We would like to stress that the main goal of the paper is not to propose new methods to account for remaining uncertainty within response surfaces, but rather to consider fuzzy thresholds when partitioning a response surface. The research question needs to be made clearer in the paper: how to divide a response function in success and failure regions, when the threshold that defines success is ambiguous?

This was exemplified for three different methods used to generate regions of success and failure when the response surface itself is uncertain.

What we proposed was how to modify each of these partitioning methods to accom-modate fuzzy thresholds. They corresponded to different assumptions about how to approach the remaining uncertainty of the response surface:

1. When assuming a continuous and probabilistic response surface: logistic regres-sion. It normally requires a binary definition of success or failure, to produce a prob-ability map. We decompose it for a sample of alpha-cuts to consider a non-binary definition.

2. When assuming a continuous response, but do not probabilities: analytical approximation. We modify the membership function with an error interval and apply it to the fitted estimates.

3. When avoiding both a continuous approximation and reliance on probabilities: convex hulls. Here there is no membership function, hulls are enlarged to consider a looser definition of success and failure.

In the revised version we will be refine the scope of the paper. We realize that the paper, even with these clarifications, attempts too many things at the same time. Testing several partitioning methods leads to confusion about the scope of the paper and detracts from the main contribution on fuzzy thresholds. We thus suggest an important departure from the first version: removing methods 2 and 3. The focus is then on introducing fuzzy thresholds to the logistic regression-based response surface.

The title, abstract, introduction and methods chapters will be revised accordingly, insisting on the following points:

1) Response surfaces as a common tool in DMDU (decision-making under deep uncertainty) normally rely on a binary definition of success and failure. In practice however, the difference between success and failure is not always clearly defined.

2) The research question is: how to use response surfaces in such cases? We propose a method to incorporate fuzzy thresholds to the response surface.

3) Doing so on a response surface that is itself well-defined is straightforward. Instead of a frontier, a clear transition area separates the regions of "full" success and "full" failure. However, we face the challenge of applying a fuzzy definition of success and failure to an uncertain surface. The notion of possibility attempts to synthesize the different nature of the uncertainties that we try to associate in a single figure for decision support: uncertainty of inputs - and thus performance - on the response side, uncertainty of the appreciation from the decision-maker or ambiguity, on the valuation side.

4) We propose a mathematical justification for an approximated logistic regression that would consider a sample of values between 0 and 1, instead of binary only.

*"However, the authors do not discuss Quinn et al. (2018), who used logistic regression as described in this paper to define success/failure regions that account for stakeholders' different levels of risk aversion by choosing different probabilities of success from the logistic regression to define the boundary. The authors also state that logistic regression cannot capture nonlinear relationships in the mapping of climate conditions to success/failure, but this is not true. One can easily incorporate interaction or nonlinear predictors in a logistic regression just as in a linear regression. See Hadjimichael et al. (2020) for an example. Other studies which use logistic regression for scenario discovery that were not cited by the authors include Lamontagne et al. (2019) and Marcos-Garcia et al. (2020)."*

Thank you for the suggested references, they will be included in the revised version. It will be interesting to discuss how stakeholders' risk aversion defines success/failure regions in Quinn et al. (2018), how fuzzy thresholds rather relate to stakeholders' ambiguity (or sometimes loss aversion), and how possibilistic surfaces attempt to combine both.

The statement concerning the lack of non-linearity between variables was indeed not correct, the revised manuscript will be changed accordingly.

*"With respect to the convex hull representation of possibilistic surfaces, this sounds like info-gap decision theory (Ben-Haim, 2006), which the authors do not discuss in the paper. It is not clear what their method contributes beyond this approach."*

We propose removing the convex hull approach from the paper. Still we here discuss the difference with the info-gap method, that the original paper should indeed have mentioned.

The info-gap method starts from a first guess coordinate in the exposure space and increases the uncertainty horizon from this estimate through a nested set of convex hulls. Their increasing size is controlled by a parameter. In this study, we assume no initial estimate, we consider the uncertainty space as a whole and draw the hulls containing all successes or failures, and we thus do not quantify the uncertainty horizon. There was also a difference in scope. We did not propose a complete framework like info-gap. We integrate fuzzy targets to different tools that are commonly employed in DMDU, which can then be used within larger frameworks like info-gap, eco-engineering decision scaling, etc.

*"It is also worth noting potential problems with this approach. One, which is briefly described by the authors, is that if the failure boundary is not convex, it could be too conservative. For example, a failure region like the red region in the attached figure could be estimated by logistic or linear regression with an interaction term between the two factors on each axis to capture the non-convexity. The convex hull, however, would include everything to the right of the black line, which includes a substantial region of successes in blue. But a convex hull might not always be more conservative like the authors imply. This is because it is defined by the realized values from their model simulations. As discussed by the authors with respect to their logistic regression model, none of the GCMs met their failure definition, but the probability of success in those worlds did not always meet their threshold of acceptability. They might not fall within the convex hull of failures, though, making the convex hull a less conservative definition in that case."*

This should indeed have been specified as a limitation to convex hulls. Risk aversion is indeed not always an advantage for the convex hull. Its main strength is rather as an alternative when no good extrapolating tool (logistic regression or direct approximation) can be used. E.g. if the stressors are poor descriptors and/or the response is too noisy, or if the decision maker just wants to rely on actual simulations only to define the

regions. The border of the hulls represents a variation in the exposure space, without having to make any statement about the distribution of the observed phenomena within or outside the hull.

As previously mentioned, the 3 methods explored how to incorporate fuzzy thresholds for different ways to handle the remaining uncertainty of the response. In the case of convex hulls, the fuzzy definitions of success and failure simply change the samples that they delineate.

*"Where I think the authors have introduced a new approach to the literature is in combining logistic or linear regression with fuzzy set membership. The question is, what value does this method add to the alternative approaches? I think this should be the focus of the paper, and it is not currently clear what that value is. Personally, I find the first and third approaches more intuitive. It is easy to understand what a probability of success represents, so defining success regions based on probability contours from a logistic regression makes sense to me. Similarly, it is easy to understand a failure region defined by lines connecting the farthest scenarios in which failures have occurred. I find fuzzy sets much harder to interpret, and more subjective to define. But I think it could provide value in that no hard success/failure threshold has to be assumed if using it with linear regression, whereas this is not true for the other two approaches. It would be helpful to expound more on this benefit, and the differences that come out of using this approach as opposed to Method 1. It is likely no method dominates all others, but why is this new method on the Pareto front of options? This needs to be better emphasized by comparing and contrasting the regions that come out of the alternative approaches."*

This is an important element, also asked by the first referee.

As previously mentioned, we were not comparing a fuzzy set approach to two other approaches. The three methods are different ways to divide an uncertain response in

success and failure regions. Fuzzy sets were meant to extend these different methods for cases where the threshold defining success and failure is ambiguous. Removing methods 2 and 3 should clarify this.

The revised manuscript will focus on the incorporation of fuzzy thresholds using only one partitioning method. What we propose is an extension of bottom-up vulnerability assessment studies for specific situations where the distinction between failure and success states are not clearly defined. This is not an alternative to traditional bottom-up approaches. The ultimate goal of the proposed extension is to be able to address a particular situation whereby crisp thresholds do not exist due to a variety of reasons including the lack of consensus amongst stakeholders, the ambiguous definition of the associated objective, etc.

This comment also asks to emphasize the differences that might come out of this approach, we agree it is worth discussing. However, the result section will remind that when comparing crisp and fuzzy thresholds, the crisp threshold is only counterfactual, not an alternative method to be compared to. It helps to visualize how fuzzy thresholds affect the partition of the exposure space in regions (which can be further emphasized at figure 11) but we assume that the crisp threshold is not available in the first place.

Similarly, it is also worth considering if fuzzy thresholds can lead to a different decision, compared to a counterfactual crisp threshold. In the attached figures (to be included in section 2 of the revised manuscript), we see that it is theoretically the case if the response functions of two options have different slopes. Other case studies show this change in slope, e.g. Quinn et al. (2017) where an improvement for expected flooding can make extreme floods worse, thus changing the slope of performance as function of stressors. However, in the present paper, the difference in slopes between the alternative options is small and does not lead to a different decision. Still, we will make clear that the results specific to this case-study are an illustration of how our proposed extension works, rather than a justification of its comparative value.

*"Finally, the authors discuss shortcomings of using gridded scenarios to build models of success/failure regions, but they never compare their non-gridded sampling to a gridded sample to illustrate its claimed superiority. I suggest the authors remove this argument entirely as it is a secondary argument anyway, and is never actually illustrated. Please see the annotated manuscript for additional, more minor comments."*

We will indeed follow the suggestion and remove the argument.

We here respond to some of the additional comments in the manuscript. We again really appreciate the time and dedication to this review, including the English corrections.

Line 5, abstract: we could say there is both noise and error, but those are hard to distinguish in the response surface approach. We will replace the sentence by: "furthermore, response surfaces have their own irreducible uncertainty, from the limited number of descriptors and the stochasticity hydro-climatic conditions".

Line 58: the proposed definition is indeed more adequate, modified.

Line 90, modified: "the choice of a longer modelling time scale. . ."

Line 95: although the logistic regression is indeed used in scenario discovery, we here focus only on how it probabilistic regions instead of binary regions in the exposure space (scenario discovery starting with this step and focusing on the transition boundary). The sentence is reworked: "Quin et al. (2018), Kim et al. (2019), Lamontagne et al. (2019), Hadjimichael et al. (2020), and Marcos-Garcia et al. (2020) use a logistic regression to divide the exposure space based on probability of success (often as a step in a scenario-discovery approach)."

As requested at line 209, and based on comments lines 128, 156, 181, 211, we need to clarify some definitions on performance, probability of success, and thresholds.

- As noted in the later comments, here p is not probability of success but performance (in this case study, reliability). Replaced by r in the manuscript for clarity (in turn, error

R replaced by $\delta$).

- The performance is the reliability, i.e. the frequency of non-flooding days in a single realization, a time series, that is a point on the response surface.

- System success and failure are defined over a time series, by the performance satisfying a threshold (we now replace "performance target" by "acceptability threshold" in all instances for clarity). They can be confused with single-time step "failures" or "successes" (flooding, or lack or flooding), so we make sure to insist in the revised manuscript. In turn when defining a state of flooding, we specify "flooding threshold".

- The logistic regression provides a probability of success, while success itself is a frequency (of local successes, i.e. non-flooding days).

Line 116, added: "overall system success or failure are measured over a time series, as opposed to local successes or failures that happen at a given time step. . ."

Line 218, modified: "Removing the noise through aggregation. . ."

Line 181: this uncertainty is not considered in the calculation, only in the discussion. We will also remind it in this section.

Line 206, added: "a sigmoid error function"

Line 215: this part will be deleted, but indeed, intuitively the actual error distribution would make more sense.

Section 3.2, lines 272-302: paragraphs restructured as recommended.

Line 306, coefficient of variation added with literature examples (Nazemi 2020).

Line 324: removed as unnecessary.

Lines 352 and 357: the un-gridded argument is removed, as suggested in the main comment.

Line 364: the McFadden pseudo R square is computed in the revised manuscript.

Line 400: sentence removed.

Line 512: sentence removed.

References:

Quinn, J. D., Reed, P. M., Giuliani, M., and Castelletti, A. (2017). Rival framings: A framework for discovering how problem formulation uncertainties shape risk management trade-offs in water resources systems, Water Resources Research, 53, 7208–7233.

Additional references to be included in the paper on membership functions:

R. E. Haber, R. Haber, A. Alique, and S. Ros (2002). Application of knowledge-based systems for supervision and control of machining processes. Handbook of software engineering and knowledge engineering, vol. 2, pp. 673-710.

J. M. Garibaldi and R. I. John (2003). Choosing membership functions of linguistic terms. The 12th IEEE International Conference on Fuzzy Systems, 2003. FUZZ '03., St Louis, MO, USA, 2003, pp. 578-583 vol.1, doi: 10.1109/FUZZ.2003.1209428

B. Bouchon-Meunier, M. Dotoli, B. Maione and D. Bari. (1996). On The Choice Of Membership Functions In A Mamdani-Type Fuzzy Controller.

Wu D. (2012). Twelve considerations in choosing between Gaussian and trapezoidal membership functions in interval type-2 fuzzy logic controllers. IEEE International Conference on Fuzzy Systems (FUZZ-IEEE); Brisbane, QLD, Australia

Sadollah, A. (2018). Introductory Chapter: Which Membership Function is Appropriate in Fuzzy System? 10.5772/intechopen.79552.

―――――――――――――

[Figure]

**Fig. 1.** With a crisp threshold $\theta$, rule 2 has a larger success region A2.

Performance r

——— Rule 1

——— Rule 2

Θ1

Θ2

stressor X

**Fig. 2.** With a fuzzy threshold ($\theta 1$, $\theta 2$), Rule 2 has a larger "at least partial" success domain S2, but a smaller "full" success domain C2, than Rule 1.